# *Right* this way: Can VLMs Guide Us to See More to Answer Questions?

**Li Liu**[*]    **Diji Yang**[*]    **Sijia Zhong**    **Kalyana Suma Sree Tholeti**
**Lei Ding**    **Yi Zhang**    **Leilani H. Gilpin** [†]
University of California, Santa Cruz
{lliu112,dyang39,szhong16,ktholeti,lding25,yiz,lgilpin}@ucsc.edu

## Abstract

In question-answering scenarios, humans can assess whether the available information is sufficient and seek additional information if necessary, rather than providing a forced answer. In contrast, Vision Language Models (VLMs) typically generate direct, one-shot responses without evaluating the sufficiency of the information. To investigate this gap, we identify a critical and challenging task in the Visual Question Answering (VQA) scenario: can VLMs indicate how to adjust an image when the visual information is insufficient to answer a question? This capability is especially valuable for assisting visually impaired individuals who often need guidance to capture images correctly. To evaluate this capability of current VLMs, we introduce a human-labeled dataset as a benchmark for this task. Additionally, we present an automated framework that generates synthetic training data by simulating "where to know" scenarios. Our empirical results show significant performance improvements in mainstream VLMs when fine-tuned with this synthetic data. This study demonstrates the potential to narrow the gap between information assessment and acquisition in VLMs, bringing their performance closer to humans. Our dataset and code are available at: https://github.com/LeoLee7/Directional_guidance.

## 1 Introduction

In recent years, Vision-Language Models (VLMs) have made significant strides in general multimodal tasks such as visual recognition and Visual Question Answering (VQA) [1, 49]. This progress has opened up a vast potential for various applications, including enhancing visual accessibility for visually impaired individuals [4, 20], supporting decision-making in autonomous systems [50, 32], enabling interactive technologies [39], etc. Despite these advances, VLMs still fall short of human capabilities. Humans can intuitively assess whether the available information is sufficient to answer a question and seek additional details when necessary [36, 11]. In contrast, VLMs typically tend to provide direct, single-response outputs even when information is insufficient to answer the question accurately. This limitation reduces their effectiveness in real-world applications [13]. To address this issue, recent studies have explored ways to teach VLMs to assess information sufficiency [42]. These studies aim to have VLMs either provide concrete answers or label questions as *unanswerable*, using benchmark datasets from real user questions like VizWiz [20].

However, a significant gap remains in handling *unanswerable* cases: deciding what actions to take when VLMs identify a question to be *unanswerable*. Humans naturally possesses the ability to seek additional details when faced with unanswerable questions — a challenge often encountered in real-world VQA tasks due to poor image quality, ambiguous questions, or loss of context [3, 9]. To

---

[*]Equal contribution.
[†]Corresponding author: lgilpin@ucsc.edu.

38th Conference on Neural Information Processing Systems (NeurIPS 2024).

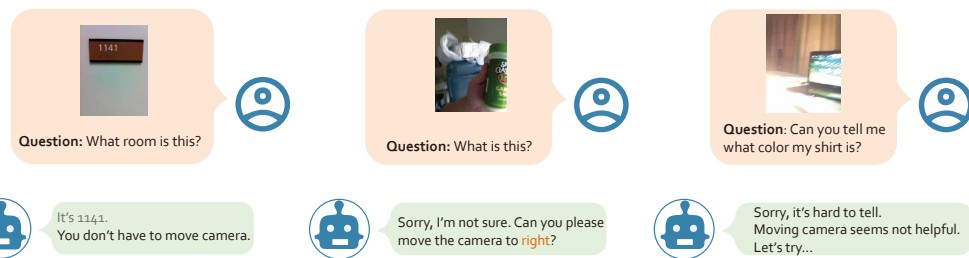

Figure 1: The examples of the Directional Guidance task. The model utilizes self-knowledge to distinguish between known and unknown information and provides guidance on where to find more information.

the best of our knowledge, no existing benchmarks focused on "what to do" after the model identifies information insufficiency. This active process of information acquisition, fundamental to human cognition, has not been replicated in VLMs and remains largely unexplored.

To narrow the gap between VLMs and human intelligence, we suggest going beyond improving accuracy on answer generation or merely deciding on information sufficiency. Instead, we focus on enhancing the model's capability to provide constructive feedback when encountering *unanswerable* questions. In response to this challenge, we introduce a novel VQA task aimed at providing Directional Guidance, which aligns with real-world needs, particularly for visually impaired individuals. As indicated in previous studies [9], a common issue is that many images taken by visually impaired users are ill-framed. Our task aims to guide users on how to reframe their images during the interactive VQA process. This task evaluates the model's ability to understand visual direction and determine a potential direction to obtain more relevant information.

Moreover, to empower VLM with such guiding capability, we propose an automatic VQA data augmentation framework. This framework begins by prompting a pretrained VLM to filter a set of *answerable* questions from the given VQA dataset. The corresponding images are then perturbed using predefined rules that crop relevant visual information, making it more challenging for the model to answer the questions correctly. Finally, the VLM is fine-tuned using this augmented dataset, with the task of providing Directional Guidance on resolving the predefined perturbations. This approach simulates information inadequacy scenarios and holds promising potential for enhancing the model's ability to guide users in acquiring relevant information.

To validate the effectiveness of the approach, we contribute a manually labeled test set containing the Directional Guidance for real-world *unanswerable* datasets with images taken by visually impaired individuals. Our experiments on three popular open-source VLMs show significant improvements in the models' performance on the Directional Guidance task after fine-tuning with our synthetic training data. Notably, the best-performing model outperforms GPT-4o (CoT) [31] by 3% accuracy score.

The contributions of this study are:

- **Directional Guidance task:** We define a novel VQA task. As shown in Figure 1, the proposed task assesses the model's ability to identify the information sufficiency and provide Directional Guidance when needed.
- **Directional Guidance dataset:** We create a human-labeled test set to benchmark the guidance-providing capability of VLMs.
- **Directional Guidance framework:** We propose a data-efficient framework for training models on the Directional Guidance task. This framework includes synthetic training data generation and model fine-tuning, which can be generalized with any VQA dataset with grounding information.

## 2    Related Work

**Directional Visual Understanding.** Many studies have identified that current VLMs struggle to interpret and understand spatial relationships within an input image, especially on fundamental visual concepts like relative directions [15, 40]. This ability is important for interactive VQA applications

like autonomous agents [18, 17], visual navigation, and assistive technologies designed for visually impaired individuals [24, 13]. To enhance VLMs' capability to understand directional relationships, researchers construct extensive training data [44], add assistive visual prompt [47, 30], or include collaborative VLMs to communicate and ensemble their decisions [8]. However, these methods often require heavy data collection or introduce additional models. Previous studies have investigated visual learning through simulation [18], but they rely on virtual interactive environments that may not accurately reflect real-world scenarios. Another trend involves generating training data by asking questions about directional relationships in existing images [29, 27], but this also requires additional involvement of advanced models. In our study, we aim to improve the model's directional understanding with simple data augmentation methods, using images collected from real users.

**Assistive technology for visually impaired individuals.** Over the past decade, applications like VizWiz [4] and Be My Eyes [13] have used real-time video connections to enhance visual accessibility. VLMs present a more accessible and responsive solution to satisfy the user's needs, as they can provide immediate responses when given a photo-query pair. However, as noted in [9], visually impaired users often face challenges in capturing clear images. In real-world conditions, many images suffer from quality issues such as blurriness, obstructions, and improper exposure, making them difficult to recognize. These issues often result in divergence in human annotations [9, 3]. Moreover, even when the images are clear, the questions may still be difficult to answer due to the off-framing of the target objects [9]. Addressing these challenges typically requires multiple rounds of queries and adjustments to properly frame the key object. For VLMs, these difficulties may be amplified because their training data typically lack examples of unrecognizable images. Additionally, to align with the multiple adjustment interaction offered by human operators in Be My Eyes applications, VLMs need to offer honest and effective guidance to navigate to target objects.

**Self-knowledge.** Self-knowledge refers to the model's ability to recognize what is known and unknown [22]. When confronted with unanswerable questions due to ambiguity or insufficient information, VLMs/LLMs often generate hallucinated responses [26, 46]. Previous research has introduced methods to help LLMs understand limitations regarding unknowns [48, 2, 34, 41]. Subsequent studies, such as [12], have explored explaining unanswerability by constructing known and unknown datasets through data augmentation and refining base models with a self-curation method. For VLMs, [25] presents a robust visual instruction tuning dataset that includes negative instructions at different semantic levels, i.e. nonexistent object manipulation, existent object manipulation, and knowledge manipulation, all implemented by GPT-4. Although these studies validate the benefits of data augmentation, they have focused on generating negative or unknown data primarily within the language modality. Instead, our study extends this exploration into multimodal data by incorporating the vision modality.

# 3 The Cognitive Question: From What's Unknown and Where to Know

To understand how a statistical model conceptualizes the world, one effective approach is to draw an analogy to human cognition. In the meta task in NLP (i.e., Question-Answering as other core NLP tasks can be transformed into QA), human cognitive processes in problem-solving and learning are multifaceted, involving not just the retrieval of stored information but also the recognition of one's knowledge boundaries and the strategic acquisition of new knowledge. To simulate these processes, we propose a hierarchical cognitive process pattern comprising three levels:

1. **Response Generation (knowing what's known):** At the foundational level, the model utilizes its existing knowledge base and basic analysis capabilities to generate responses to queries. This process mirrors the human cognitive function of retrieving known information from memory, akin to recall or recognition tasks in cognitive psychology[28, 35]. It reflects the model's ability to combine available information into coherent answers.

2. **Awareness of Knowledge Limits (knowing what's unknown):** The second level reflects the model's metacognitive ability to evaluate its own knowledge state, recognizing when it lacks sufficient information to answer a question accurately [14, 36, 37]. This awareness is crucial for intellectual honesty and mirrors the human cognitive process of monitoring and evaluating one's understanding and capabilities, a key aspect of metacognition [51, 38, 11].

3. **Knowledge Acquisition Direction (knowing where to know the unknown):** At the most advanced level, the model identifies pathways for acquiring new knowledge when existing

information is insufficient. This ability to seek out and engage in learning opportunities mirrors the human cognitive strategies for addressing knowledge gaps, such as identifying resources, formulating questions, or modifying learning strategies. It signifies the model's capacity for self-guided learning and adaptation, similar to strategic learning and problem-solving in human cognition.

As mentioned in Section 1, most existing works on VLMs cognitive questions are focused on the first two levels [42, 15, 5], and the third level is mostly under-explored. We argue that the challenges lie in the difficulty of collecting suitable data for benchmark and training data: there are few VQA samples that exhibit both awareness of knowledge limits and knowledge acquisition direction. Therefore, in our study, we focus on benchmark dataset curation and training data generation.

## 4 Method

### 4.1 Directional Guidance Task

We define our Directional Guidance task as follows: in the context of VQA, given an image-question pair $< I, Q >$, the model $M$ should determine whether the image needs to be reframed. To be specific, if the target object is only partially visible and not sufficient to answer the question, the model should give clear guidance for the reframing direction (left, right, up, or down). Otherwise, the model should inform whether the question is already answerable (no need to change) or remains unanswerable even with potential reframing (none of the other options). This task mirrors real-world scenarios where visually impaired individuals need guidance to position their cameras correctly through many attempts. Although the target object might be only partially visible on each attempt, with continuous adjustments under guidance, the user can always capture a better view and finally have a better chance to get the question answered. This task goes beyond simply detecting the ill-framing issue of the image: it assesses whether the framing issue impacts the model's ability to answer the specific question posed. For example, reframing may not be necessary if the question can be directly answered with the available visual information even if the image is ill-framed. We regard these guide responses as an additional output that complements the original VQA answering process.

This task exemplifies three levels of the hierarchical cognitive pattern discussed in Section 3. Instead of a binary classification of answerable/unanswerable as proposed in [9], this task emphasizes the model's ability to effectively utilize available visual information. It requires the model to assess what is known and determine where to acquire extra information, standing in the transition from unanswerable to answerable.

### 4.2 Directional Guidance Dataset

**Benchmark dataset.** To evaluate model performance in our task setting, we created a benchmark dataset derived from VizWiz dataset families [20, 7]. The VizWiz dataset consists of real VQA queries collected from visually impaired individuals [20]. From this dataset, we used all the *unanswerable* samples (1.4k) from the validation set as the training set may have potential leakage issues during the pre-training process of VLMs. We invited 26 human annotators to identify ill-framed photos and label the most promising direction to move the camera, by which the reframing action could potentially help to answer the question (more details are available in Appendix A.1). After cleaning and re-evaluation, we collected 291 samples where reframing could potentially lead to an answer, and 230 samples unlikely to be answered even with reframing. The rest samples are where the human annotators have disagreements. The details of the data collection are presented in the Supplementary Materials. To ensure a balanced distribution in the test set, we randomly selected 300 samples from the VizWiz-grounding test set and simulated the case where the current image already has sufficient information to answer the question, under the assumption that the visual evidence could be theoretically grounded in the image. Combining those three groups, we get a high-quality Directional Guidance benchmark dataset including 821 samples. Despite the size of the dataset being relatively small, this reflects the inherent challenge of the task, where ill-framed images are rare in standard VQA datasets but commonly seen in real-world scenarios. Furthermore, our dataset's diversity and comprehensiveness make it suitable for evaluating model performance on the target task, providing a valuable foundation for future studies.

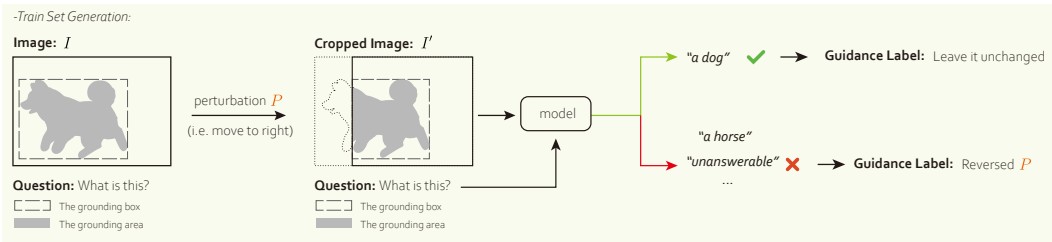

Figure 2: The training set generation framework.

**Training dataset.** We propose a data augmentation process to simulate the ill-framed samples, instead of collecting ad-hoc images that suit the task. Initially, we take all the training samples from a dataset pool - the validation set of VizWiz-grounding dataset [7] as it includes the visual groundings for each answerable VQA query. With that visual grounding information, manual perturbations have been applied to simulate ill-framing. Specifically, we identify the bounding box surrounding the target object and divide it into 10 zones, horizontally and vertically. We then choose a specific zone for cropping, resulting in an image that has some missing information while retaining a part of the target object. With a series of perturbations, we observe the consistency of the model's response to the initial VQA query and capture the cases where an ill-framing issue impacts the question-answering. As the VizWiz is an open-ended task, we use precision as the evaluation:

$$\text{Precision } = \frac{\sum_w |P(w) \cap T(w)|}{\sum_w |P(w)|} \tag{1}$$

$P(w)$ and $T(w)$ denote a word from the model prediction and from the ground-true answer. Precision calculates how many words in the predictions also appear in the ground-truth answer, and we set a threshold $e$ to identify the correctness. Following [19], only non-stop words have been taken into consideration. Figure 2 and algorithm 1 outline the process of generating training data with guidance labels.

Another crucial case in the benchmark test set involves samples that remain unanswered even after adjusting the camera. One more data argumentation technique has been placed: we mismatch the questions and images from the same dataset pool to create new pairs with different semantic information. Most questions in the original dataset pool are generic, as a highly frequent question is "What is this?" without semantic information. Correspondingly, our GPT-4 enabled argumentation helped rephrase the paired question and answer. For example, given an image $I_i$ with the question $Q_i$ "What is this?" and an answer $A_i$ "laptop," the new question $Q_i'$ will be rephrased to "What's the color of this laptop?." Then, we mismatch the $Q_i'$ with another irrelevant image $I_j$ to form a new pair. This augmentation generates complex, real-world queries where straightforward answers are infeasible, compelling models to learn deeper semantic information.

---

**Algorithm 1:** Generate synthetic training set with data augmentation

**Input:** A set of image-question pairs $(I, Q)$
**Output:** A dataset $D$ with perturbed image-question pairs and their corresponding guidance

Initialize dataset $D$ as empty
Define a range of perturbation magnitudes $\mathcal{P}$ with types {left, right, up, down}
**foreach** *pair* $(I, Q)$ **do**
 **if** *model $M$ predicts $(I, Q)$ correctly* **then**
  **foreach** *perturbation $P$ in $\mathcal{P}$* **do**
   $I' \leftarrow$ Apply $P$ to $I$
   **if** *$M$ still correctly predicts $(I', Q)$* **then**
    $G \leftarrow$ 'leave it unchanged'
   **else**
    $P_{\text{reverse}} \leftarrow$ Reverse operation of $P$
    $G \leftarrow P_{\text{reverse}}$
   Add $(I', Q, G)$ to $D$

---

## 4.3 Experiment settings

**Model Selection.** To verify the feasibility and effectiveness of our approach for different model architectures and sizes, we analyze the experiments of four mainstream open-source large models with different sizes, including: LLaVA-1.5 [27], InstructBlip [10], GPT-4o [31], and CLIP [33]. First, we benchmark the test set on the LLaVA-1.5, InstructBlip, and GPT-4o on the zero-shot setting. A series of prompts has been designed to test their zero-shot performances, serving as our baselines. Next, we generate a training dataset using algorithm 1 and apply LoRA [21] fine-tuning on the open-sourced models. We anticipate the effectiveness of our proposed training framework will be reflected by the improvement of model performance compared with the zero-shot baseline.

**Task format.** To quantitatively analyze the model's ability to provide guidance, we format the task with a basic VQA multiple choice template: "`<image>{Original_Question}` `To improve the image and answer the question, how should the camera be moved? A.Leave it unchanged. B.Left. C.Right. D.Up. E.Down. F.None of the other options.`" Each option reflects the model's decision of Directional Guidance: The `leave it unchanged` option indicates that the current image contains all the necessary information to answer the question. The four directional options suggest that the relevant object is only partially visible, and further image adjustment is needed. The `None of the other options` implies that moving the camera will not help because the question is inherently unanswerable, i.e. due to the ambiguity, or the relevant object is absent from the current image. We use the F1 score and accuracy as the evaluation metrics and also analyze the confusion matrix of the different options.

**Zero-shot prompt setting.** For the zero-shot baseline, we enhance the basic template with additional instructions and explanations tailored for each model. We designed two prompt settings to accommodate their varying capabilities. The first setting is a single-round query where the model makes predictions from six options directly. The second setting is a two-round prompt, following the Chain-of-Thought [43] process. This two-round prompt decomposes the tasks and works as follows: Initially, we prompt the model to determine if the target object is fully present in the image, partially visible, or if the question is unanswerable. The corresponding options are: `leave it unchanged`, `reframe`, and `none of the other options`. If the model indicates that the target object is only partially visible, we then ask it to decide a specific direction for movement: `left`, `right`, `up`, or `down`. To ensure reproducibility, we include all prompts we used in Supplementary Materials A.5.

**Fine-tune setting.** In our training framework, we utilize data augmentation to generate potential samples with guidance labels. We assess the consistency of the model's predictions before and after perturbations and categorize the samples into two groups. Samples where the model fails to predict post-perturbation are considered positive, and their Directional Guidance labels are assigned one of four directions: `left`, `right`, `up`, or `down`. Conversely, samples where the model maintains correct predictions are labeled as negative, with the Directional Guidance label set to `leave it unchanged`. Upon analyzing these groups, we observed that negative samples predominated the generated training set. To ensure a balanced distribution within the training dataset, we under-sampled the negative samples to align with the average count of the four directional categories. We also adjusted the number of `None of the other options` samples to achieve an even distribution across the entire training set.

After generating the training set, we format the new pairs into a standardized instruction fine-tuning layout: each sample, comprising $< I', question >$, is supplemented with option choices and instructions. Since the task requires models to respond with a single letter, the prediction process is equivalent to a classification task. Following the settings in [27], the loss is only computed on the token for the chosen letter and the $< eos >$ (end of the sentence) token. Also, to prevent the model from memorizing the letter distribution, we randomly shuffle the association between letters and options, ensuring each letter (from A to F) is paired with an option randomly in each training sample. When fine-tuning each model, most training configurations follow the officially suggested settings, and more training details are presented in Supplementary Materials A.2.

**None-generative Models.** Since the task has been simplified to a classification problem, we also investigate whether a non-generative model with a simpler architecture could suffice. Accordingly, we add a linear probe layer onto CLIP and perform a classification head, using a vision encoder (CLIP-ViT-L-336px) aligned with LLaVA-1.5 and a text encoder in the default setting. We concatenate the image feature and the text feature as the input for the classification head. Since the CLIP model can not generate open-ended answers, we use the synthetic training dataset generated by LLaVA-1.5 13b.

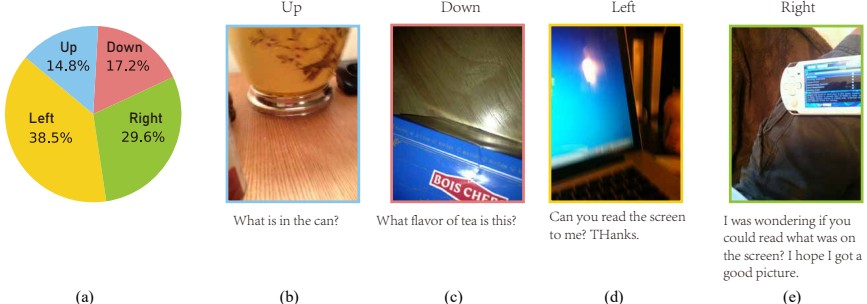

(a)  (b)  (c)  (d)  (e)

Figure 3: The distribution of four directions in our benchmark dataset (a) and examples (b-e). The upper caption is the Directional Guidance label and the lower caption is the original question.

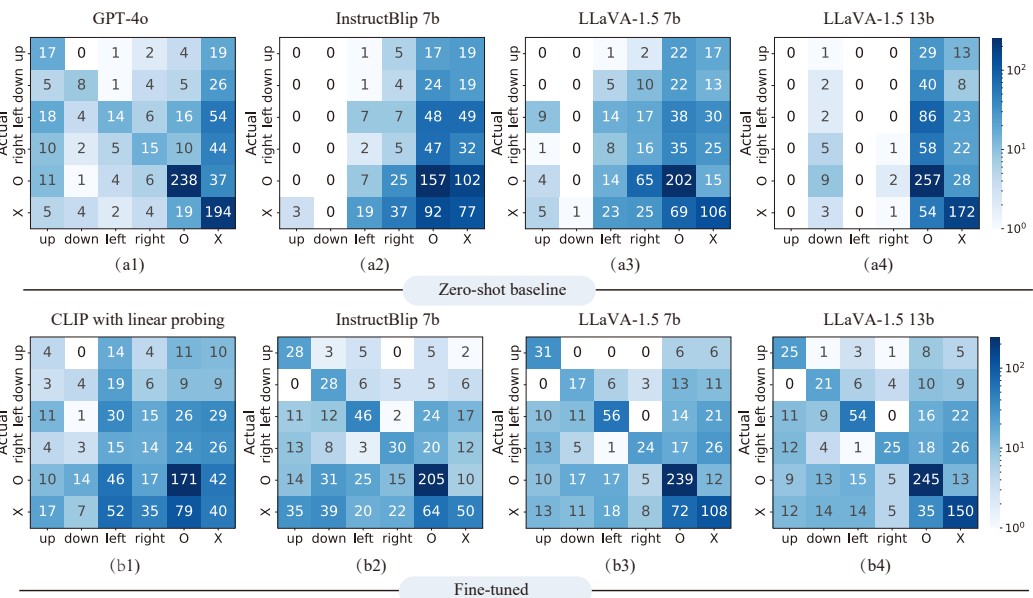

Figure 4: The heatmaps of the model's prediction. (a1)-(a4) shows the baseline performance under zero-shot setting, and (b1)-(b4) shows the performances of fine-tuned models. 'O' denotes the class `leave it unchanged`, and 'X' denotes the class `none of the other options`.

## 5 Results

### 5.1 Directional Guidance benchmark dataset and baseline performance

Fig. 3 (a) shows the distribution of four directions in our benchmark dataset. The horizontal directions are the most common, with left at 38.5% and right at 29.6%. The figure displays four typical samples from each direction. We also identified a frequent scenario where users need to take another photo and attempt a different question, as shown in Fig. 3 (e). This pattern reveals a common challenge for visually impaired individuals: without continuous guidance, the user and assistant can easily lose the context of the original VQA. These findings emphasize the importance of providing clear, sequential dialogue-based guidance for effectively adjusting the camera position.

As mentioned in Section 4.3, we use different prompt settings for each group of models that suit their capabilities. For the 7b models, we use the two-round prompt because these models benefit from a more structured, step-by-step approach, which helps them handle the task more effectively. In contrast, we tested the LLaVA-1.5 13b and GPT-4o model with a single-round of prompting to see if they were capable of this task. The prediction results are presented in Fig. 4, from (a1) - (a4). We observe that three open-sourced models (LLaVA-1.5 7b/13b, and Instructblip 7b) show similar behaviors: these models tend to avoid predicting the reframing cases and mistakenly categorize them

as either `leave it unchanged` or `none of the other options`. With the two-round prompt, LLaVA-1.5 7b and InstructBlip 7b make some correct predictions in the `left` and `right` categories. However, the correct and incorrect predictions are nearly balanced, with frequent misclassifications in the opposite direction. For example, the number of true `left` predictions is equivalent to the number of erroneous `left` predictions that were intended to be `right`. For GPT-4o, there are fewer errors in categorizing reframing cases into the wrong categories, and more cases within the reframing category are correctly predicted. However, contradictory predictions also occur frequently within the reframing cases. The result demonstrates that all the models are generally incapable of accurately predicting the reframing cases under zero-shot prompt settings. However, every baseline model performs well in the `none of the other options` category. We present the examples from each model in the Supplementary material A.4 to visualize the model's pretrained capability on the Directional Guidance Task.

## 5.2   Model's performance after fine-tuning

We present the heat maps of the fine-tuned model predictions in Fig. 4. By comparing the prediction results between the zero-shot baselines and the fine-tuned models, we observe significant and consistent improvements in prediction performance, demonstrating the effectiveness and generalizability of our proposed method. Although there is considerable potential to improve the overall accuracy, the fine-tuned models reduce confusion between `reframing`, `leaving it unchanged`(O), and `none of the other options`(X). The fine-tuned models are more likely to provide directional guidance on the reframing cases. Moreover, the predictions within reframing cases show noticeable improvement, as indicated by the clear diagonal line in the heat map. Another interesting finding is the substantial reduction in wrong predictions with opposite directions (e.g., predicting an up case as `down`). This clarity is meaningful, as it lowers the chance of users receiving conflicting guidance, thereby enhancing safety and efficiency in real-world applications. Overall, the fine-tuned models reduce errors across all options, showing significant improvement in both cross-category and within reframing predictions.

To evaluate our training framework's sensitivity to different settings, we conducted groups of comprehensive experiments. We used three metrics to quantitatively assess the model's performance: overall F1 score, overall Accuracy, and Accuracy on the reframing cases denoted as ACC(F). The metrics for the baseline models are presented in Table 1. In some baseline experiments, we found that the zero-shot setting did not always ensure a standard output format. In such cases, we performed post-processing and excluded samples with predictions that did not fall within our options. The total number of excluded samples was fewer than 10, and this only occurred in the zero-shot baseline models. The results, as shown in Table 1, include a combination of different settings from two aspects: varying perturbation ranges and the impact of shuffling letters and options in the training data. Regarding the choice to shuffle, we observed that randomly mixing letters and options does not consistently enhance performance. For instance, with a perturbation range of 0.1-0.9, the unshuffled approach often outperformed the shuffled version, while with a perturbation range of 0.3-0.7, shuffling generally resulted in inferior performance.

The CLIP with linear probing method achieves comparable performance with the zero-shot performance of InstructBlip, but it's still not able to provide informative guidance (the accuracy for random choice for a six classification task is 16.6%). This suggests that simple CLIP-based encoders, lacking integration with a language model, may not be sufficient for this task. While the task largely relies on the model's perception of salient features, the contextual information within questions is also essential. For instance, the VizWiz dataset includes many generic questions such as 'What is the color of this?' Many of these questions are answerable even though the photos are heavily ill-framed. Since these questions do not concern spatial details, the appropriate guidance is to `leave it unchanged`. This underlines a key distinction between our task and other image quality detection tasks, especially those focusing on ill-framing solely on image modality.

## 6   Discussion

In this section, we analyze the effect of different settings, including perturbation range and shuffling operations, on the generation of training data. A detailed heatmap of the model's predictions is presented in Supplementary Materials with Figure 6. The perturbation range determines the crop

Table 1: Model's performance with different settings. F1 and ACC denote the F1 score and accuracy score, respectively. ACC(F) refers to the accuracy of reframing directions, excluding the categories 'Leave it unchanged' and 'None of the other options.'N/A indicates not applicable experiments due to limitations on the model's accessibility or incompatibility with the experiment design.

| | Shuffling | | W/O shuffled options | | | | | | With shuffled options | | | | | | Baseline | | |
|---|---|---|---|---|---|---|---|---|---|---|---|---|---|---|---|---|---|
| | Perturbation range | | 0.3-0.7 | | | 0.1-0.9 | | | 0.3-0.7 | | | 0.1-0.9 | | | | | |
| | Metrics | | F1 | ACC | ACC(F) | F1 | ACC | ACC(F) | F1 | ACC | ACC(F) | F1 | ACC | ACC(F) | F1 | ACC | ACC(F) |
| Model | CLIP+linear probe | | 0.30 | 0.31 | 0.19 | 0.32 | 0.32 | 0.18 | | | | | N/A | | | | |
| | InstructBlip-7b | | 0.40 | 0.41 | 0.35 | 0.46 | 0.47 | 0.45 | 0.43 | 0.42 | 0.35 | 0.45 | 0.46 | 0.27 | 0.27 | 0.31 | 0.04 |
| | LLaVA1.5-7b | | 0.47 | 0.48 | 0.33 | 0.57 | 0.58 | 0.44 | 0.55 | 0.56 | 0.36 | 0.52 | 0.54 | 0.31 | 0.39 | 0.42 | 0.10 |
| | LLaVA1.5-13b | | 0.54 | 0.54 | 0.41 | **0.63** | **0.63** | 0.43 | 0.55 | 0.54 | **0.52** | 0.56 | 0.57 | 0.37 | 0.43 | 0.53 | 0.01 |
| | GPT-4v | | | | | | | N/A | | | | | | | 0.52 | 0.59 | 0.13 |
| | GPT-4o | | | | | | | | | | | | | | 0.56 | 0.60 | 0.19 |

ratios used to generate the training samples, ranging from 0.1 (minimal crop) to 0.9 (maximum crop). We observed that positive Guidance samples tend to cluster at high crop ratios, while lower ratios often correspond to negative samples (where the Guidance is `leave it unchanged`). Therefore, a range of 0.3-0.7 leads to a more balanced selection of training data, while a range of 0.1-0.9 provides a more comprehensive and varied dataset. This setting can affect the model's performance due to the balance between the diversity and complexity of the generated training samples, and the trade-off works as follows: when the perturbation becomes more severe (e.g., at a ratio of 0.9), images are aggressively corrupted. This increases the chance of obtaining positive Guidance samples, as the model is more likely to fail in predicting these heavily perturbed samples, which it could have predicted accurately without perturbation. However, it also results in significant information loss and greater challenges in detecting objects, making it a harder sample to learn. Conversely, a moderate perturbation ratio results in less aggressive cropping, allowing the model to access more information and better respond to the original question. However, this can lead to fewer positive Guidance samples, as the perturbation does not sufficiently challenge the predictions. The differences with the shuffling settings could be attributed to the regularization effect, which prevents the model from memorizing fixed patterns and increases training difficulty, especially when training data is scarce. In scenarios with less training data, shuffling acts as a form of data augmentation, increasing the diversity of training examples and making the model more robust. However, shuffling might add unnecessary complexity to a larger training dataset derived from a wider perturbation range, making it harder for the model to learn effectively. In such cases, the unshuffled approach allows the model to quickly identify and leverage consistent patterns, facilitating faster and more efficient learning processes.

**Ablation Study**  To gain a more comprehensive understanding of how different perturbation ranges affect model performance, we conducted a more fine-grained ablation study with LLaVA1.5-7b. Specifically, we evaluated perturbation ranges of 0.1-0.3, 0.3-0.5, 0.5-0.7, and 0.7-0.9, alongside our main experiments of 0.1-0.9 and 0.3-0.7. The results, presented in Table 2, reveal that as the perturbation range increases from 0.1-0.3 to 0.5-0.7, both overall F1 scores and overall accuracy show substantial improvements, stabilizing around 0.49. However, at the highest perturbation range of 0.7-0.9, we observe a slight decrease in the ACC(F) metric, suggesting that overly aggressive perturbations may introduce excessive complexity, hindering the model's ability to accurately identify relevant objects. Notably, the broader range of 0.1-0.9 achieves the highest overall F1 and accuracy scores, suggesting that a wide perturbation range strikes an effective balance between data diversity and sample complexity. Additionally, all perturbation ranges demonstrate improvements in ACC(F), with enhanced reframing direction performance as perturbation increases, except at the highest range. These findings support our initial discussion by emphasizing the trade-off between data diversity and the complexity of perturbed samples. Future work could explore optimized strategies for selecting perturbation ranges, potentially employing dynamic or adaptive methods to further improve model performance based on specific dataset characteristics.

Table 2: Model performance under different perturbation ranges with LLaVA-1.5 7b.

| Perturbation Range | Overall F1 | Overall Accuracy | ACC(F) |
|---|---|---|---|
| 0.1-0.3 | 0.19 | 0.24 | 0.31 |
| 0.3-0.5 | 0.49 | 0.49 | 0.38 |
| 0.5-0.7 | 0.49 | 0.49 | 0.43 |
| 0.7-0.9 | 0.50 | 0.49 | 0.40 |

# 7 Limitation and Future Work

In this study, we focused specifically on guiding image reframing directions as a proof of concept. Although reframing is one of the most common needs when assisting visually impaired individuals, some other aspects that impact the VQA process could also be explored, such as orientation, exposure, and focus. Our data augmentation framework can be extended to these aspects and generate training data in a similar way, and we plan to explore this in future work. Second, the directional guidance has been simplified to a classification task on directions, which may not fully capture the complexity of real-world scenarios. For example, effective reframing might require combining multiple directions—such as moving both up and left—or even zooming out. A more informative guidance in practice would also consider additional parameters like the magnitude of the reframing action. Those complexities can confuse the model, leading to inaccurate evaluations. To enhance clarity and reduce ambiguity in our benchmark dataset, we included only cases that received consistent annotations from multiple annotators, which resulted in a limited size in our benchmark test set. Additionally, our preliminary experiments are designed to validate the effectiveness of our proposed framework rather than to maximize the model's performance. Consequently, the current method cannot fully guarantee the reliability of the model's prediction, and it still requires cautious deployment in high-risk scenarios. Moving forward, we aim to refine the task design and data generation framework, adapting more effectively to complex, real-world applications. In addition, theoretically, our guidance framework can be extended to more general and quantitative scenarios. By simulating spatial drift, we can customize the ratio of drift and produce synthetic training data with quantitative values. This potential extension would allow models to identify not just the direction but also the extent of camera movement required. This makes such quantitative guidance particularly meaningful in applications such as robotics, where precise tracking of target objects is crucial, for example, in calculating the ground truth for the extent of movement needed [16]. Furthermore, we expect our unsupervised data generation framework to alleviate the pressing data needs of studies exploring LLM or VLM for spatial or temporal reasoning tasks [6, 45, 23].

# 8 Conclusion

In this paper, we introduced a novel task and benchmark dataset within the context of Visual Question Answering (VQA) aimed at improving the self-knowledge of Vision-Language Models (VLMs). Our task specifically evaluates how well VLMs can assess the sufficiency of visual information and determine the necessary actions to reframe an image to obtain additional information. To address the challenge of limited training data, we proposed an automated framework that generates synthetic data by simulating unanswerable scenarios through perturbations applied to answerable cases. Our results show that current high-performing VLMs, including LLaVA and GPT-4o, struggle with this task, revealing a gap in their ability to handle incomplete or ambiguous visual inputs. However, when fine-tuned with the synthetic training data generated by our framework, the models significantly outperformed the zero-shot baseline on real-world data.

This study highlights the importance of self-knowledge in VLMs, particularly their ability to recognize the boundaries of available information and take appropriate actions when confronted with incomplete or misleading data. By mimicking human cognitive processes, our approach presents a promising solution for enhancing models' self-knowledge and robustness, especially in real-world applications that require accurate and adaptive responses. This is particularly relevant for assistive technologies, such as those designed for visually impaired individuals, where providing timely and effective guidance is essential. As VLMs continue to evolve, the ability to recognize their knowledge boundaries and make informed decisions will be critical for their successful deployment in dynamic environments. Future work can explore additional strategies for refining this self-knowledge, potentially leading to more robust models capable of learning in complex, uncertain scenarios.

# 9 Acknowledgment

We sincerely appreciate all collaborators who contributed to this project for their invaluable insights and support. Special thanks to our human labelers for their diligent efforts in creating the benchmark dataset. We are also grateful to Prof. David Lee and Prof. James Davis for their expert guidance, constructive feedback, and encouragement throughout this research.

We thank the VizWiz community for providing an inspiring platform that significantly influenced our work. We acknowledge the use of the VizWiz VQA and VizWiz Grounding datasets(both under CC-BY 4.0)). Our model development was based on LLaVA (Apache-2.0 license) and InstructBLIP (in compliance with the license terms of LLaMA and Vicuna).

This material is based upon work supported by the Air Force Office of Scientific Research under award number FA9550-24-1-0149.

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

# A Appendix / supplementary material

## A.1 Benchmark Dataset Collection

We invited 26 human annotators to label the 1.4k *unanswerable* samples extracted from the validation set of VizWiz VQA dataset[20]. We divided the dataset evenly, assigning each annotator a segment to work on. Each sample has an image and corresponding question from the original dataset. We include several sub-tasks for each VQA question to categorize how the images could be improved to better answer the question: annotators were required to choose options from the following categories— `Reframing`, `Other Actions`, or `No Way to Answer`. In the `Reframing` category, annotators specified the direction — `left`, `right`, `up`, `down`—for a reframing action that might reveal an answer. The 'Other Actions' includes the actions that are beyond the previous four actions, such as zoom in/out, rotation, and adjusting exposure. The 'No Way to Answer' category is used to indicate cases where there is unlikely to yield additional information by taking any actions. We also asked the annotators to summarize their selected options with one sentence, which can be open-ended and qualitative comments. This is used for sanity checks and helps our further cleaning. After the initial round of annotations, we engaged four additional annotators (validators) to review the labels, noting any errors or disagreements. These validators conducted two rounds of evaluation and discussion before finalizing the benchmark dataset. After the validation, we select the `Reframing` and 'No Way to Answer' categories to join our benchmark dataset. The instructions given to annotators are:

---

**Task Overview**

Thank you for joining this task! In Visual-Question-Answering (VQA), some image-question pairs are marked "unanswerable" due to insufficient information in the image. Our goal is to determine if specific camera adjustments or other actions could potentially make these questions answerable.

For each image-question pair, your task is to identify if any adjustments or guidance could potentially make the question answerable. Please select the most applicable option from the drop-down cells and provide a brief explanation in the summary column. Below is a description of each option.

**Reframing:** Choose "`Left, Right, Up`", or "`Down`" if moving the camera in a specific direction could reveal information to answer the question. Choose "`Leave it unchanged`" if the current image already contains all the information needed to answer the question.

**Other Actions:** Select "`Zoom in/out, Rotate`", or "`Adjust exposure`" if these camera settings might improve visibility or clarify the image content.

**No Way to Answer:** Choose this option if no adjustment would help, such as cases with lost context, ambiguity, or obstructions.

**Summary Column:** In the summary column, provide a one-sentence explanation of your choice to clarify the reasoning behind your selection.

---

Table 3: Instruction to annotators.

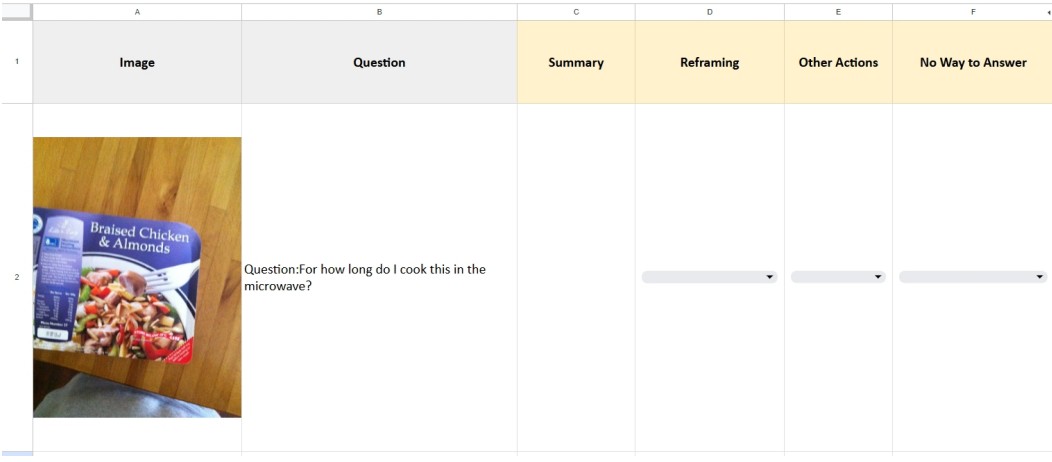

Figure 5: A screenshot of the annotation work.

## A.2 Training details

We run experiments on an Ubuntu 22.04.4 LTS system equipped with 4*NVIDIA RTX A6000 GPU and AMD Ryzen 24-Cores CPU.

The training details for LLaVA models follow the default settings in [27], with a training epoch of 3.

The training details for InstructBlip-7b are shown as follows:

| Hyper-parameters | Value |
|---|---|
| learning rate | 5e-5 |
| LoRA r | 128 |
| LoRA alpha | 256 |
| batch size | 4 |
| training epoch | 5 |

## A.3 The model's performance under different settings

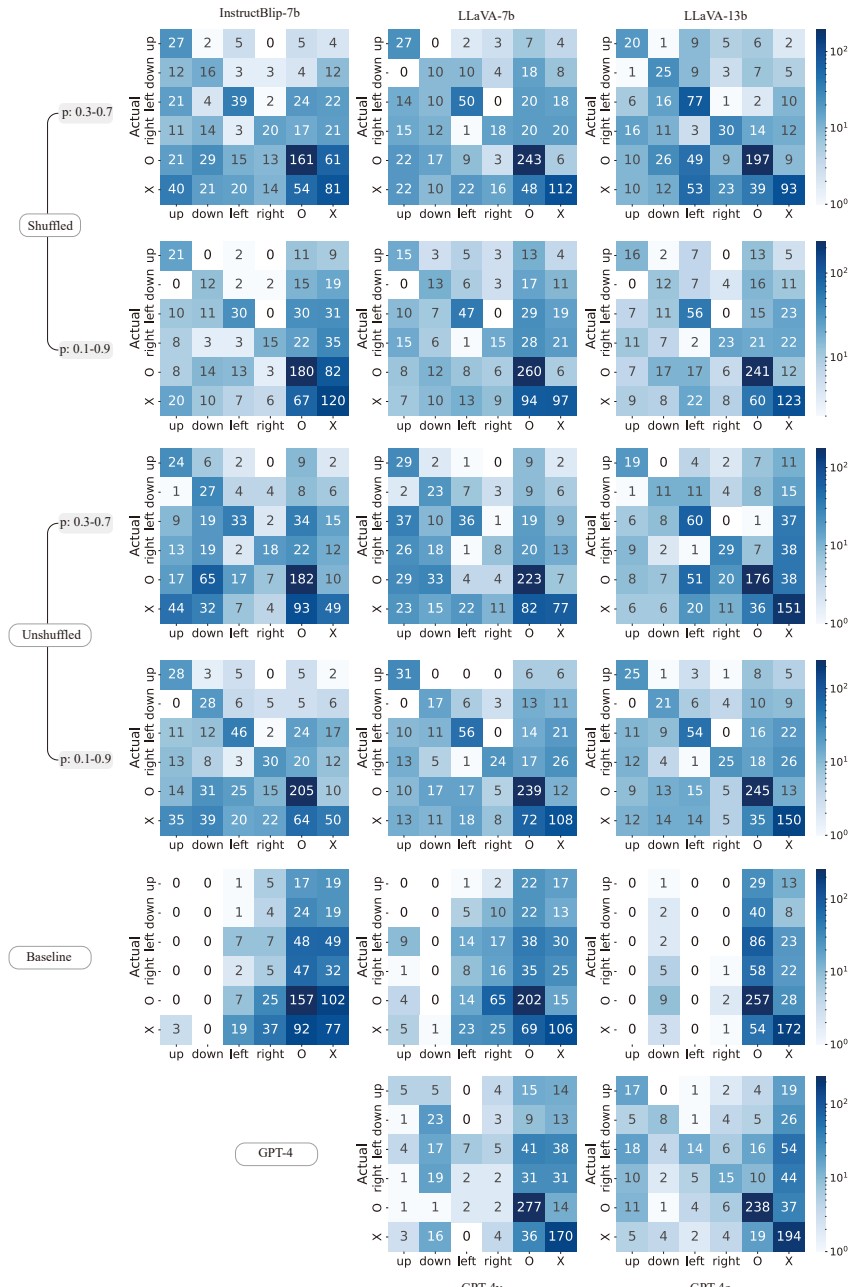

Figure 6: The heatmap of model's predictions under different settings. The $p$ denotes the perturbation ranges we applied when generating synthetic data.

## A.4 Examples of VLMs performances with zero-shot settings

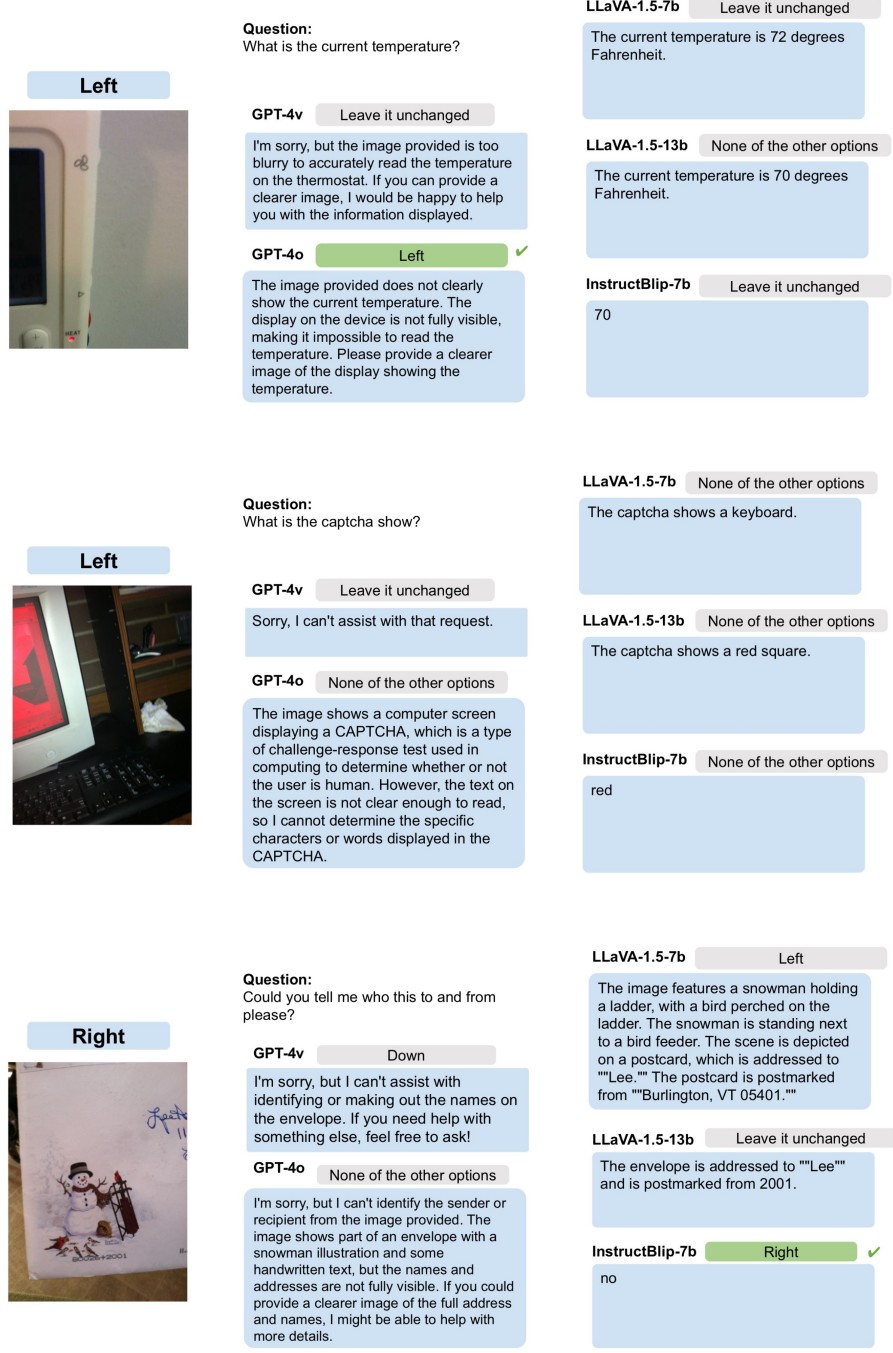

Figure 7: The examples of the VLMs' zero-shot performance on our Directional Guidance benchmark dataset.

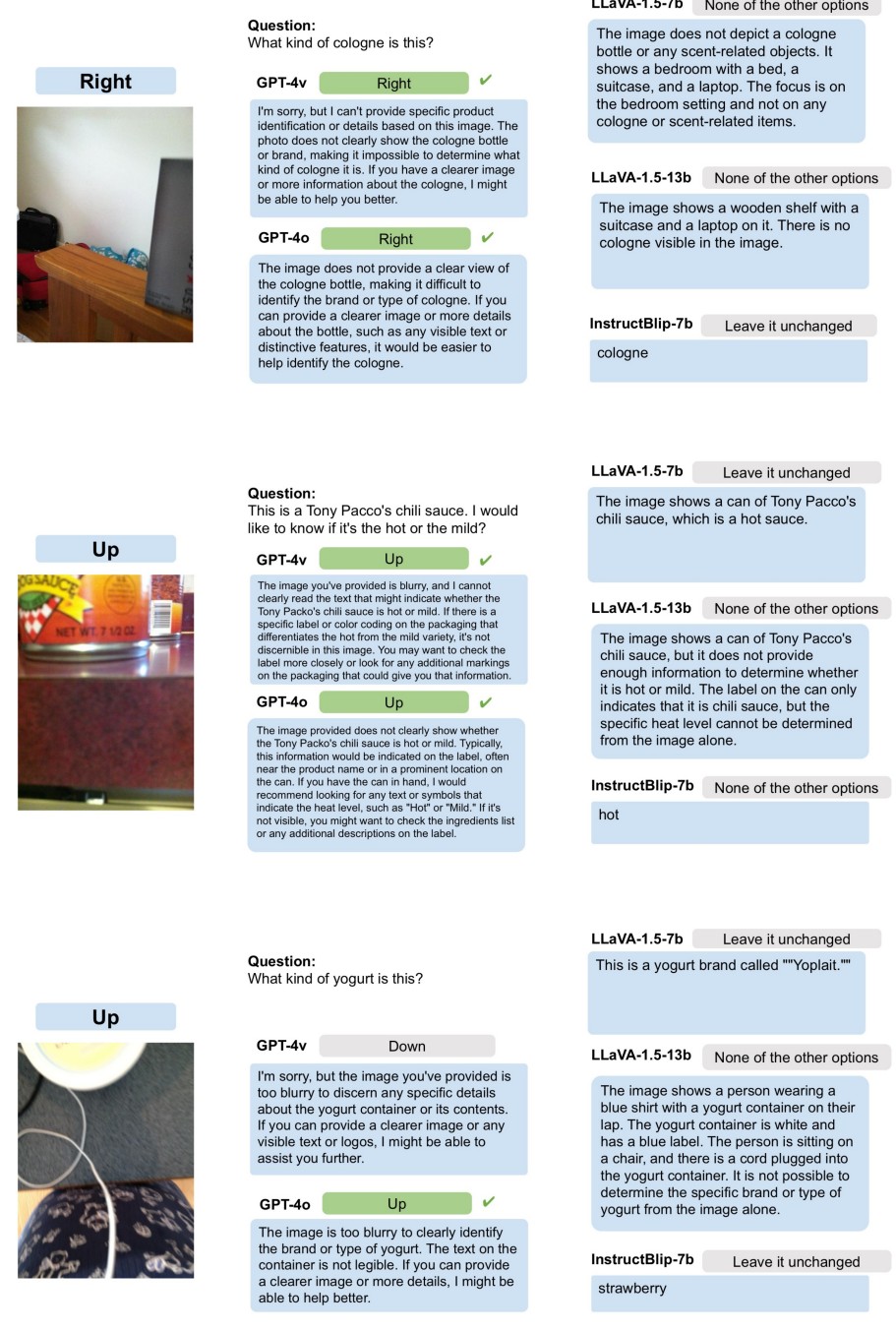

Figure 8: The examples of the VLMs' zero-shot performance on our Directional Guidance benchmark dataset.

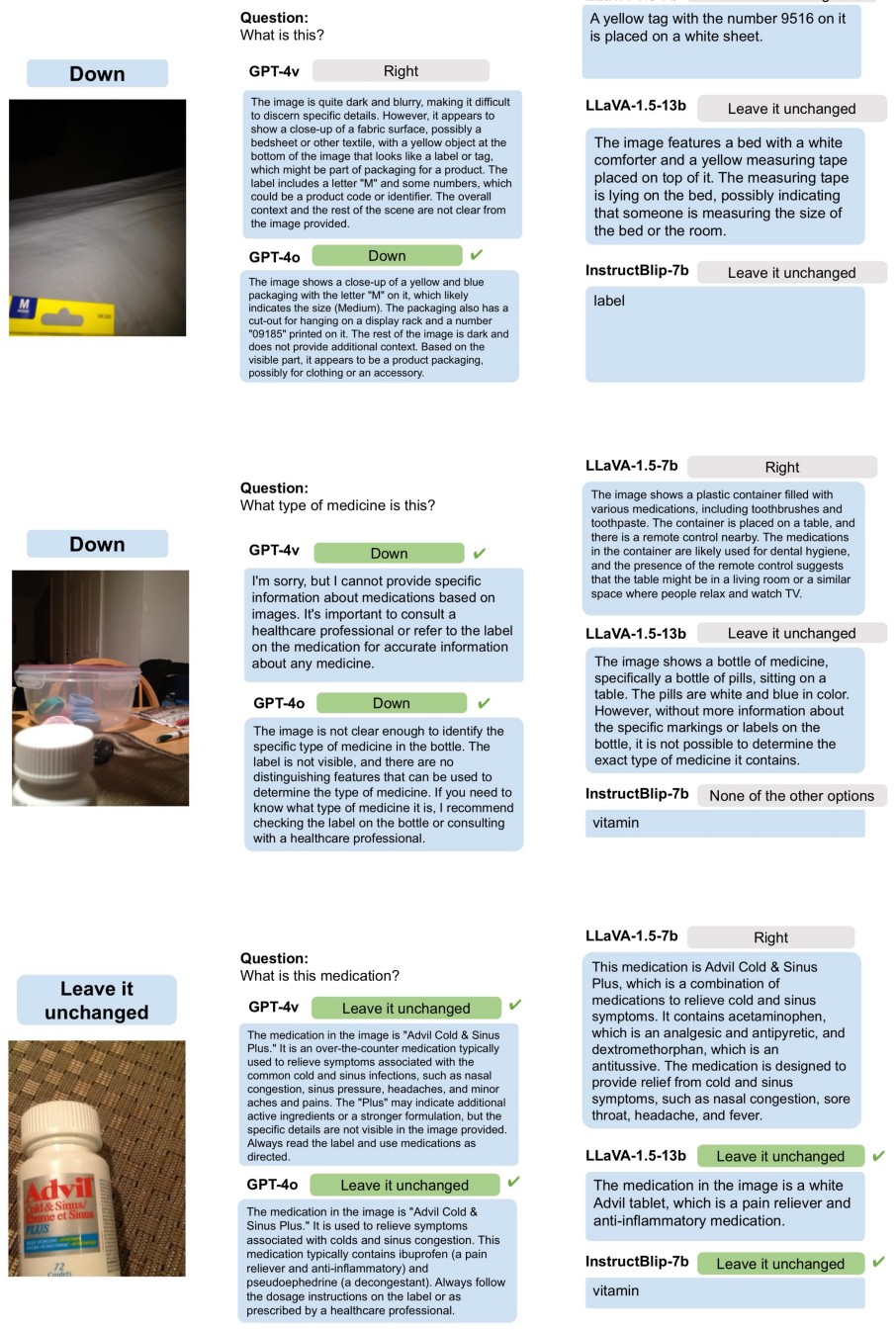

Figure 9: The examples of the VLMs' zero-shot performance on our Directional Guidance benchmark dataset.

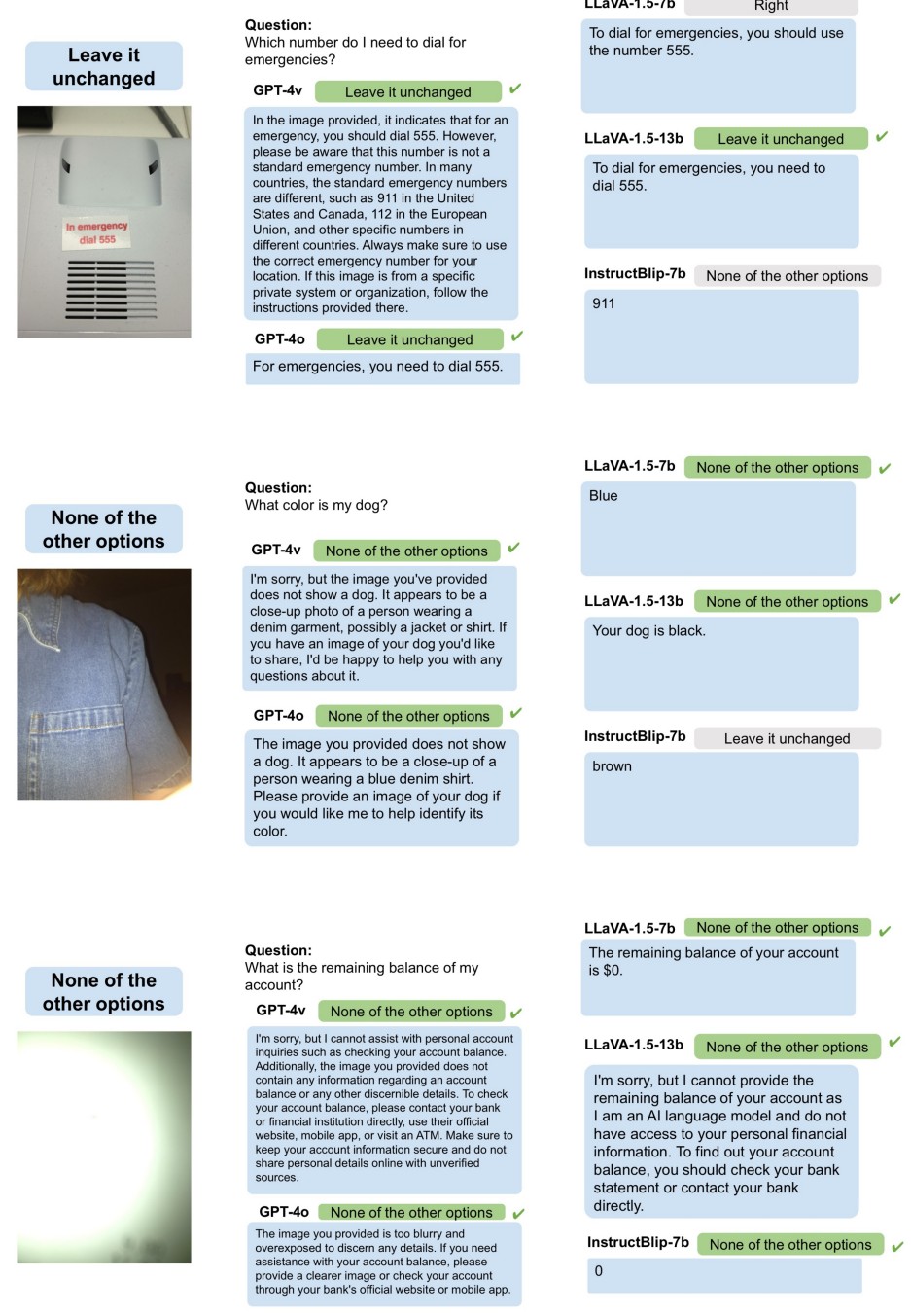

Figure 10: The examples of the VLMs' zero-shot performance on our Directional Guidance benchmark dataset.

## A.5 Prompts used for baselines

—————— INSTRUCTBLIP-7B: TWO-ROUND PROMPT —————-

**Round 1**

"'Image question pair
Image: <Image>
Question: {*QUESTION*}'"

You're an assistant who helps us adjust the view of the image to better answer the given question. Given the image and question pair, your options are as follows:

A: 'framing' - If the given image contains partial needed information and the camera can move in a certain direction to gather more information for better answering the given question.

B: 'leave it unchanged' - if the image contains enough information to answer the given question, or if the given question is already answered, choose this option.

C: 'none of the other options' - If the previous options don't fit, or the given question is not related to the given image, or the given question cannot be answered, choose this option.

- If the given question is unrelated to the given image, select C: 'none of the other options'.

- If there is no way to use the given image for answering the given question, select C: 'none of the other options'.

Please only output the selected option for the given image and question pair.

A short answer to the question is:

**Round 2**

"'Image question pair
Image: <Image>
Question: {*QUESTION*}'"

In the previous setting, the result is framing, which means that the image has part of the needed information for answering, and the camera can move to the corresponding direction to better answer the question.

You're an assistant who helps us move the camera in direction to better answer the given question. Given the image and question pair, your options are as follows:

D: 'left' - if the image is truncated on the left and the camera framing should be moved left to better answer the question based on the given image, choose this option.

E: 'right' - if the image is truncated on the right and the camera framing should be moved right to better answer the question based on the given image, choose this option.

F: 'up' - if the image is truncated on top and the camera framing should be moved up to better answer the question based on the given image, choose this option.

G: 'down' - if the image is truncated on the bottom and the camera framing should be moved down to better answer the question based on the given image, choose this option.

Please only output the selected option for direction.

A short answer to the question is:

Table 4: Prompt for InstructBlip-7b.

————— GPT4: SINGLE-ROUND PROMPT —————

You are an assistive technology specializing in visual question answering, i.e., the task of providing a natural language answer to a question about a given image. To better answer the question {*QUESTION*} based on the given image, please choose one of the six options (A. 'Leave it unchanged', B. 'Up', C. 'Left', D. 'Right', E. 'Down', F. 'None of the other options') on the camera framing. The definitions of each of the options are given below:

- A: 'Leave it unchanged' - The question can be answered based on the given image without the need for changing camera framing and there is visible text or complete object in the image to answer the question. The image is clear and shows the object in question without any truncation or need for reframing. The entire object is visible and identifiable.

- B: 'None of the other options' - The question cannot be answered based on the given image, even with a change in camera framing, or the question seems to be unrelated to the content of the image provided, or the question is incomplete or the question is unrelated to the image or there is no visible text on the image to answer the question.

If the answer to the question, i.e., visible text or object is partially visible in the image or the specific text content is not clear due to the angle and quality of the image or the piece of text is partially obscured, and the necessary details to answer the question are not discernible, please choose one of the below camera framing accordingly.

- C: 'Up' - The camera framing should be moved up to better answer the question based on the given image.

- D: 'Left' - The camera framing should be moved left to better answer the question based on the given image.

- E: 'Right' - The camera framing should be moved right to better answer the question based on the given image.

- F: 'Down' - The camera framing should be moved down to better answer the question based on the given image.

Output format required is:

- Option and its value.

————— GPT4: TWO-ROUND PROMPT —————

**Round 1**

You are an assistive technology specializing in visual question answering, i.e., the task of providing a natural language answer to a question about a given image. To better answer the question {*QUESTION*} based on the given image, please choose one of the given options (A. 'Leave it unchanged', B. 'None of the other options', C. 'Move camera') on the camera framing. The definitions of each of the options are given below:

- A: 'Leave it unchanged' - The question can be answered based on the given image without the need for changing camera framing and there is visible text or complete object in the image to answer the question. The image is clear and shows the object in question without any truncation or need for reframing. The entire object is visible and identifiable.

- B: 'None of the other options' - The question cannot be answered based on the given image, even with a change in camera framing, or the question seems to be unrelated to the content of the image provided, or the question is incomplete or the question is unrelated to the image or there is no visible text on the image to answer the question.

- C: 'Move camera' - If the answer to the question i.e., visible text or object is partially visible in the image or the specific text content is not clear due to the angle and quality of the image or the piece of text is partially obscured, and the necessary details to answer the question are not discernible.

Output format required is:

- Option and its value,
- Reason for choosing that option,
- If option A 'Leave it unchanged' is selected, give answer to question based on the given image.

**Round 2**

You are an assistive technology specializing in visual question answering, i.e., the task of providing a natural language answer to a question about a given image. To better answer the question {*QUESTION*} based on the given image or previous context *{RESULT}*, please choose one of the given four options (A. 'Up', B. 'Left', C. 'Down', D. 'Right') on the camera framing. The definitions of each of the options are given below:

If the answer to the question i.e., visible text or object is partially visible in the image or the specific text content is not clear due to the angle and quality of the image or the piece of text is partially obscured, and the necessary details to answer the question are not discernible, please choose one of the below camera framing accordingly.

- A: 'Up' - The camera framing should be moved up to better answer the question based on the given image.

- B: 'Left' - The camera framing should be moved left to better answer the question based on the given image.

- C: 'Down' - The camera framing should be moved down to better answer the question based on the given image.

- D: 'Right' - The camera framing should be moved right to better answer the question based on the given image.

Output format required is:

- Option and its value,
- Reason for choosing that option.

Table 5: Prompt for GPT4.

———————— **LLAVA-1.5-13b: SINGLE-ROUND PROMPT** —————-
USER: <image>

*{QUESTION}*

To improve the image and answer the above question, how should the framing should be changed in order to answer the question? Please select the most suitable ways to adjust the framing based on the following description:

Select 'none of the other options' if there is no correct way to change the framing for answering the above question, which includes: the question is not related to the image or the question has grammar error. If the Question is a sentence instead of a real question, select 'none of the other options'.
Select 'leave it unchanged' if the above question can be answered without adjusting the framing and the question has no related with the image, otherwise choose 'none of the other options'.
If the question is neither 'none of the other options' or 'leave it unchanged', and if the image has part of the needed information for answering and the camera can move to the corresponding direction to better answer the question, choose the framing option which you believe is the most suitable in order to answer the question above.

Options:

left
up
down
right

Please double check to make sure that you are sure about your answer and make sure that you give same priorities to each option, and give me the option only.

ASSISTANT:

———————— **LLaVA-1.5-7b TWO-ROUND PROMPT**—————-
**Round 1**
USER: <image>

*{QUESTION}*

In order to improve the image to answer the given question, please choose which category this question-image pair belongs to and the priorities of each option are the same: 'none of the other options' or 'framing' or 'leave it unchanged'. The definitions are given below:

Select 'none of the other options' if there is no correct direction to move the camera to answer the above question, which includes: the question is unrelated to the image, or the question has a grammar error.
Select 'framing' if the image has part of the needed information for answering and the camera can move to the corresponding direction to better answer the question.
Select 'leave it unchanged' if the image contains enough information for answering the above question without moving the camera to better answer the question.
- If the question is unrelated to the image, select 'none of the other options'.
- If there is no answering the question, select 'none of the other options'.

Please answer the selected category above and double check the answer. Make sure that you choose 'none of the other options' and 'framing' and 'leave it unchanged' based on their definitions and the priority of these options are the same.

ASSISTANT:

Output format required is:

- Option and its value,
- Reason for choosing that option,
- If option A 'Leave it unchanged' is selected, give answer to question based on the given image.

**Round 2**

USER: <image>

*{QUESTION}*

In the previous setting, the result is 'framing' which means that the image has part of the needed information for answering, and the camera can move to the corresponding direction to better answer the question. Please choose the most suitable one of the four options for moving camera for better answering the question by the directions: left, right, up, down. The definitions are given below:

left - if the image is truncated on left and the camera framing should be moved left to better answer the question based on the given image.

right - if the image is truncated on right and the camera framing should be moved right to better answer the question based on the given image.

up - if the image is truncated on top and the camera framing should be moved up to better answer the question based on the given image.

down - if the image is truncated on down and the camera framing should be moved down to better answer the question based on the given image.

Answer the selected direction only.

ASSISTANT:

Table 6: Prompt for LLaVA-1.5.

