# OpenReview forum: "Right this way: Can VLMs Guide Us to See More to Answer Questions?"
_NeurIPS.cc/2024/Conference — NeurIPS 2024 poster_

### Official Review · Reviewer_SnhN · 2024-07-03

**Soundness:** 3
**Presentation:** 3
**Contribution:** 3
**Rating:** 6
**Confidence:** 3

**Summary:**

VLMs offer an interesting opportunity to describe their uncertainty if the images given to them are of poor quality. This paper explores a framework for estimating whether a VLM can identify if its ability to correctly answer a visual question would improve were it given a better picture. The paper introduces a new labeled dataset for VQA where questions involve identifying how best to take better pictures. The paper introduces also a finetuning scheme with synthetically generated examples leveraging the labeled dataset.

**Strengths:**

- The problem is interesting, relevant, and important
- The novel data set is an important contribution after it is released
- The performance improvement after fine-tuning on LLaVA is impressive.

**Weaknesses:**

- It would be interesting to see how the model performs with different prompting techniques. Beyond vanilla CoT, can we try methods covered in (https://arxiv.org/pdf/2402.07927)
- The problem specifically focuses on qualitative spatial reasoning with respect to moving the camera. Can this methodology be used for quantitative reasoning (e.g., Spatial-VLM, Spatial-RGPT), by identifying not just which direction, but by how much to move the camera?
- The two perturbation ranges of 0.1-0.9 and 0.3-0.7 are confusing. What exactly are you differentiating with these categories? It would have been interesting to evaluate over disparate ranges (e.g., 0.1-0.3, 0.3-0.5, 0.5-0.7, …) so that we can see how performance drops with the perturbations

**Questions:**

- Why is LLaVA-1.5 13b worse than 7b?

**Limitations:**

Limitations are lightly discussed

---

> ### Author Rebuttal · Authors · 2024-08-07
>
> Thank you for recognizing the unique contribution of our study and the effectiveness of our training method. We believe these advancements highlight our work's potential to drive further research.
>
> We are glad to discuss the following points:
>
> ***W1. It would be interesting to see other method Beyond vanilla CoT***
>
> **Response:**
> We agree that exploring diverse CoT methods for this task is meaningful. In our study, we experimented with a series of prompting techniques and presented the ones with the best performance based on our experiments. We will add more failure cases in the appendix in our next version to provide a comprehensive view of the challenges faced. As suggested by reviewer 1, we also made a complementary experiment with more complex zero-shot prompting techniques with the self-reflection mechanism. However, we also found that a more complex prompt does not guarantee a better performance, and that remains an interesting question.
> As prompt design remains an open question, we acknowledge that our settings may not fully tap the model's potential to achieve the best performance. Our primary aim was to test the model's baseline performance under general prompt settings, but we appreciate the suggestion to investigate further prompting methods as outlined in the referenced paper. We plan to explore these advanced techniques in future work to benchmark the current VLMs in a more comprehensive way.
>
> ***W2. Can this methodology be used for quantitative reasoning (e.g., Spatial-VLM, Spatial-RGPT), by identifying not just which direction, but by how much to move the camera?***
>
> **Response:**
> We acknowledge that our current methodology focuses on qualitative spatial reasoning and recognize this as a limitation.
>
> Theoretically, our guidance framework can be extended to more general and quantitative scenarios. By simulating spatial drift, we can customize the ratio of drift and produce synthetic training data with quantitative values. This potential extension would allow models to identify not just the direction but also the extent of camera movement required. We plan to explore this in future work.
>
> In our current task, we focused on assisting visually impaired individuals, which informed our data and question design. We assumed it would be challenging for users to accurately move the camera by a precise quantitative value. However, we agree that quantitative guidance would be particularly meaningful in applications such as robotics, where precise tracking of target objects is crucial. One challenge lies in calculating the ground-truth for the extent of movement needed. We are actively working on this direction, leveraging simulated environments such as Ge, Yunhao, et al. "BEHAVIOR Vision Suite: Customizable Dataset Generation via Simulation, CVPR 2024."
>
> ***W3. The two perturbation ranges of 0.1-0.9 and 0.3-0.7 are confusing. What exactly are you differentiating with these categories? It would have been interesting to evaluate over disparate ranges (e.g., 0.1-0.3, 0.3-0.5, 0.5-0.7, …) so that we can see how performance drops with the perturbations***
>
> **Response:**
> We understand the confusion regarding the perturbation ranges. In our manuscript, we included a discussion on the rationale behind choosing the ranges of 0.1-0.9 and 0.3-0.7. (with * in the table below)
>
> These ranges were selected to balance between image quality and sufficient data generation. Severe perturbations (e.g., 0.9) can lead to significant information loss, making the images harder to interpret, while minimal perturbations (0.1) do not challenge the model enough, resulting in fewer positive guidance data points.. To explore the effect of different perturbation range settings, we present the supplementary experiment results below,
>
>
> LLaVA 7B Performance Across Different Perturbation Ranges
>
> | Perturbation Range | Overall F1 | Overall Accuracy | ACC(F) |
> |:------------------:|:----------:|:----------------:|:------:|
> |      0.1-0.3       |    0.19    |       0.24       |  0.31  |
> |      0.3-0.5       |    0.49    |       0.49       |  0.38  |
> |      0.5-0.7       |    0.49    |       0.49       |  0.43  |
> |      0.7-0.9       |    0.50    |       0.49       |  0.40  |
> |      *0.1-0.9       |    0.57    |       0.58       |  0.44  |
> |      *0.3-0.7       |    0.47    |       0.48       |  0.33  |
>
> These findings generally align with our discussion: as the value of the perturbation range increases, the overall accuracy and F1 score improve quickly and stay relatively stable around 0.49. However, if we focus on the accuracy  on four reframing directions(ACC(F)), it improves as the perturbation range increases but drops when reaching the highest range (0.7-0.9). This demonstrates there might be a tradeoff between the diversity of directional training data and the complexity of perturbed samples. A broader range (0.1-0.9) may lead to a balance and generally good performance across metrics. However, we can obeserve a general improvement with all perturbation ranges on the ACC(F). We acknowledge that studying the optimized way to choose from different ranges of perturbations could provide additional insights into performance variations. This is an interesting direction for future work, and we will consider a more detailed analysis in our next version.
>
> ***Question:Why is LLaVA-1.5 13b worse than 7b?***
>
> **Response:**
> Thank you for your question. We believe the difference might be due to the sensitivity of prompt settings, especially in a zero-shot scenario. Interestingly, we found that the larger model, LLaVA-1.5 13b, performed worse with a two-round prompt compared to a one-round prompt, highlighting the instability of prompt settings for different models.  We believe this could be a valuable finding and points to the need for further research to understand and optimize prompting strategies across model sizes.

---

> > ### Comment · Reviewer_SnhN · 2024-08-12
> > **Thanks for the rebuttal**
> >
> > I have read the rebuttal and appreciate the author's clarifications, especially with regard to the perturbation ranges. I encourage the authors to include this discussion as well as some discussion on the effect of better prompting, CoT, etc. on the results already found.

---

> > > ### Author Response · Authors · 2024-08-13
> > > **Thanks for the response**
> > >
> > > Thank you for your feedback. Could you kindly clarify whether you are suggesting revisions to the paper's discussion section, or if you would prefer us to engage in further discussion during the current reviewer-author discussion period? We plan to include our discussion on general perturbation ranges, diverse prompting and CoT strategies in the paper's discussion section in the next revision. We will also provide more details in the supplementary materials to demonstrate the sensitivity observed through our extensive prompt explorations.

---

> > > > ### Comment · Reviewer_SnhN · 2024-08-14
> > > > **Clarification**
> > > >
> > > > To clarify, I meant to encourage your current plans on adding discussion to perturbation ranges, prompting, CoT, etc. I have no further questions. Thanks again for your thoughtful rebuttal!

---

### Official Review · Reviewer_Eiqe · 2024-07-12

**Soundness:** 4
**Presentation:** 3
**Contribution:** 3
**Rating:** 6
**Confidence:** 4

**Summary:**

- The paper proposes a recourse for unanswerable visual questions — rather than simply abstain, can a VLM indicate how to adjust an image so that the question becomes answerable?
- The authors introduce the Directional Guidance VQA task and a dataset for the task.
- The authors propose a data generation pipeline to produce data that can train models for the directional guidance task.

**Strengths:**

- The problem studied by the paper is practical and has the potential to have a real-world impact.
- There have been a number of works on VQA models that can abstain, but abstention alone is not ideal for real-world deployments because it would be frustrating to be told that your question was unanswerable.
- Providing a remedy for unanswerable visual questions via directional feedback is a useful and novel contribution.
- The framing of the task is simple and reasonable.
- The data generation pipeline is clever and uses a model-in-the-loop technique to systematically generate training instances.

**Weaknesses:**

Since this is somewhat of a resource paper, it would be nice to have multiple vision-language models of different sizes. This is also a bit of a concern given the fact that a model-in-the-loop technique is used to create a training dataset for the task, since different models would ostensibly have different weaknesses.


Another weakness is that this is a very narrowly scoped task and dataset that is maybe a better fit for a CV conference.

**Questions:**

- Table 1 is confusing and looks aesthetically unpleasant.
- Your table numbers are wrong. Ex: in L295 you refer to Table 6, but this is actually Table 1 and Section 6. Your `\label` is in the wrong place.

**Limitations:**

Not applicable.

---

> ### Author Rebuttal · Authors · 2024-08-07
>
> We are grateful for your recognition of the contribution and potential impact of our proposed study, and that's exactly what we are targeting. We also expect to facilitate many meaningful real-world applications such as AI assistants for visually impaired individuals.
>
> We appreciate the opportunity to discuss the following points.
>
> ***W1. Since this is somewhat of a resource paper, it would be nice to have multiple vision-language models of different sizes. This is also a bit of a concern given the fact that a model-in-the-loop technique is used to create a training dataset for the task, since different models would ostensibly have different weaknesses.***
>
> **Response:** To validate our training pipeline, we conducted a proof-of-concept using two mainstream open-source VLMs of different sizes. This initial approach was aimed at demonstrating the feasibility and effectiveness of our method across different model architectures and sizes.
>
> We share your concern regarding the model-in-the-loop technique, particularly the potential for the training data to be influenced by the specific weaknesses of the models used. This is indeed similar to unsolved challenges in active learning, where the quality of training data is dependent on the model's performance during data collection.
>
> Moving forward, our work will focus on exploring methods to enhance the stability and robustness of our framework. This could include regularization techniques to mitigate biases introduced by model-specific weaknesses and integrating multiple models to diversify the training data.
>
> ***W2. Another weakness is that this is a very narrowly scoped task and dataset that is maybe a better fit for a CV conference.***
>
> **Response:** Thank you for your feedback. While our study starts from a specific scenario, it is both important and generalizable. This new task fills the gap in current research regarding the handling of unknown cases and explores the capability of VLMs to acquire additional information when visual information is insufficient. Moreover, our proposed Directional Guidance Framework can be adapted and potentially applied to other tasks/modalities such as understanding query intention and reducing hallucination, where we must deal with insufficient or unclear information. These demonstrate a broader applicability of our work beyond a specific task.
>
> We appreciate your acknowledgment of the importance and uniqueness of our study in the strength part. This guidance capability is crucial when deploying vision-language models in real-world applications, not only for specific applications, including assisting visually impaired individuals, but also for advancing the general understanding of multi-modal AI systems. Therefore, we believe the core question of whether VLMs can guide us in acquiring more information is fundamental for broader AI communities.
>
> ***Q1:Table 1 is confusing and looks aesthetically unpleasant.***
>
> **Response:** Thank you for your feedback. We will edit the table in the next version to avoid confusion. For example, we will consider splitting the table into two or more parts to make the information clearer and easier to navigate, and more consistent shading will be used to highlight different sections of the table.
>
> ***Q2: Your table numbers are wrong. Ex: in L295 you refer to Table 6, but this is actually Table 1 and Section 6. Your `\label` is in the wrong place.***
>
> **Response**: Thank you for pointing out the issue with the table numbers and labels. We have reviewed and corrected the placement of the `\label` command to ensure that the tables are referenced correctly.

---

> ### Comment · Reviewer_Eiqe · 2024-08-13
>
> I have read the author response. I will maintain my rating of a 6. I do not think the paper should be rated lower because it introduces a resource annotated by humans that addresses a real problem. I do not think the paper should be rated higher because the scope is very narrow and there is little technical contribution — it's an application paper (train VLM on dataset with an handcrafted augmentation technique that works only for this application), albeit a useful one. Also, the organization and presentation of the paper could have been better: it was hard to parse conclusions from the body of the results section.

---

> > ### Author Response · Authors · 2024-08-13
> > **Thank you for the feedback**
> >
> > Thank you for your feedback and for maintaining a positive rating. We appreciate your recognition of the value our resource brings. To improve the clarity and organization of the results section, we will split out the conclusions into a separate section and enhance the overall structure to ensure our findings are more clearly presented in the next revision.

---

### Official Review · Reviewer_eg4t · 2024-07-13

**Soundness:** 3
**Presentation:** 2
**Contribution:** 3
**Rating:** 5
**Confidence:** 3

**Summary:**

This paper evaluates and expands the self-reflection capabilities of multiple MLLMs, so that MLLMs can actively evaluate the adequacy of given query information and give suggestions on how to find more reliable information. This paper proposes a hierarchical cognitive process pattern theory and creates a manually annotated dataset for evaluation.

**Strengths:**

Methods for active reflection and evaluation of given information are meaningful for enhancing the quality of MLLM responses.

**Weaknesses:**

1. The rationality of the proposed hierarchical cognitive process pattern, which comprises three levels, merits validation. First, are these three levels both necessary and sufficient for the cognitive process? If the issues at these three levels are addressed, can we then resolve the problems associated with the MLLM cognitive process? Second, does the evaluation benchmark and the ability to expand MLLM, as presented in this paper, align with these three levels?

2. The method proposed in this paper is fundamentally similar to the MLLM self-reflection method. Can it be directly implemented through self-reflection without requiring additional training?

3. A comparable method to the one presented in this paper is DualFocus [1], which not only actively evaluates image quality but also performs more sophisticated active grounding.

4. Lastly, the evaluation dataset used in this paper is relatively small, which makes the evaluation results less convincing.

[1]Cao Y, Zhang P, Dong X, et al. DualFocus: Integrating Macro and Micro Perspectives in Multi-modal Large Language Models[J]. arXiv preprint arXiv:2402.14767, 2024.

**Questions:**

Evaluate the difference and experimental effect between this method and the large model self-reflection method.

**Limitations:**

Yes.

---

> ### Author Rebuttal · Authors · 2024-08-07
>
> Thank you for acknowledging the significance of active reflection and evaluation methods. We would like to highlight that our main contribution is not only improving response quality but also providing guidance to acquire more information.
>
> We are glad to discuss the following points:
>
> ***W1. The alignment of the proposed hierarchical cognitive process***
>
> **Response:**
> Thank you for questioning the hierarchical cognitive process pattern we proposed. We appreciate the opportunity to clarify the rationale behind it.
>
> The three levels—Response Generation, Awareness of Knowledge Limits, and Knowledge Acquisition Direction—are analogies to study a model's behavior in a human cognitive way. Several studies in the field, as mentioned in our related works, evaluate model intelligence in a similar way. Our work aims to advance models from mere self-knowledge to actively seeking additional information when needed. Understanding the real cognitive process of MLLMs remains an open question. However, we believe this analogical perspective can inspire us to understand and improve MLLM performances.
>
> In our proposed task: If a model can identify the information sufficiency and provide a relevant response, it reflects the first level (knowing what's known). If the model can identify when the available visual  information is insufficient to answer the question, it demonstrates the second layer (knowing what's unknown). Finally, if the model can suggest a direction to answer the question, it exhibits a sense of knowledge acquisition direction (knowing where to know the unknown).
>
> ***W2. Comparison with training free self-reflection approach***
>
> **Response:**
> Thank you for your feedback. We do not consider our method to share significant similarities with the self-reflection approach. As you mentioned, the self-reflection method can be training-free. In contrast, our main method explores generating synthetic training data to improve the model's performance, which is fundamentally different.
>
> However, we agree that exploring self-reflection in our task would be meaningful. Currently, we use zero-shot CoT to explore the model’s baseline performance, and training-free self-reflection can be an additional way to reflect the model’s capability.
>
> However, the existing self-reflection/self-critic approaches focus on LLMs in the LLM-agent or tool-learning domains. With MLLMs, self-reflection is still underexplored and still lacks a unified or fixed way. Therefore, we made a complementary experiment based on some relevant self-reflection papers. (See response to Q1).
>
> ***W3. Comparison with DualFocus [1]***
>
> **Response:**
> We would like to emphasize the significant difference between the two works. Specifically, our task is in a scenario where the required information may not be present or only partially present in the image. Instead of focusing on locating answerable cases, our task aims to develop VLMs' ability to guide users to seek additional information to answer questions. This capability is crucial for handling unanswerable cases.
>
> We believe that both works address different challenges. We appreciate the reviewer's suggestion and will put DualFocus in our related work.
>
> ***W4. The size of evaluation dataset***
>
> **Response:**
> We appreciate the reviewer's concern on the size of our evaluation dataset. There are several points we would like to clarify in this regard:
>
> Real-World Data Collection and Cleaning: Our dataset was collected from real-world scenarios and underwent multiple rounds of rigorous data cleaning to ensure its quality. The nature of the Directional Guidance Task, which involves identifying cases with insufficient visual information, inherently results in a smaller dataset. This is because many real-world VQA datasets contain primarily high-quality, well-framed images, and the occurrence of low-quality or ill-framed cases is relatively rare. Also, to the best of our knowledge, there is no existing benchmark dataset that supports evaluating the Directional Guidance ability.
>
> Diverse Guidance Types: Despite the smaller size, we have carefully composed different groups of guidance types to maintain the comprehensiveness of our dataset. This diversity ensures that our dataset is sufficient to evaluate the model's performance across various scenarios relevant to our task. Each example in our dataset is thoughtfully chosen to represent a unique challenge in guiding the acquisition of additional information.
>
> We also draw attention to similar work in the field, such as the paper “https://arxiv.org/abs/2404.12390” where the evaluation dataset sizes are of a comparable scale. This precedent underscores that meaningful and rigorous evaluations can still be conducted with datasets of this size.
>
> In future work, we plan to expand our dataset further to include more diverse scenarios. However, we believe that our current dataset, despite its size, provides a valid foundation for evaluating our proposed task.
>
> ***Q1: Evaluate the difference and experimental effect between this method and the large model self-reflection method.***
>
> **Response:**
> Continuing from W2, we implemented a self-reflection approach by constantly reassessing the current information and incorporating the image and all conversational contexts. It includes three steps: identifying answerability, finding the key object, and suggesting proper directional guidance. Each step involves local self-reflection to confirm the information, and interact with adjacent steps until reaching an initial answer. Finally, we integrate all conversations and self-reflect the answer with the global context.
>
> We tested this self-reflection pipeline with GPT-4o-mini. With a series of carefully designed self-reflection pipeline and prompt templates, the F1, ACC, and ACC(F) are 0.53, 0.53, and 0.23 correspondingly, which is comparable to our reported baseline. As a reference, our best performing model achieves 0.63, 0.63, 0.43.

---

> > ### Comment · Reviewer_eg4t · 2024-08-13
> >
> > Thanks for the authors' detailed response. Their reply effectively addressed my concerns, so I decide to increase the score from 3 to 5.

---

> > > ### Author Response · Authors · 2024-08-13
> > > **Thank you for the response**
> > >
> > > Thank you for the thoughtful reconsideration of our work. We're glad that our explanations addressed the previous concerns, and we appreciate your decision to adjust the score based on our clarifications.

---

### Author Rebuttal · Authors · 2024-08-07

We thank the reviewers for their time and valuable feedback. We are encouraged that the reviewers recognize our contribution in threefold:

- This work identifies a meaningful problem with real-world impact: "Can VLMs guide us in acquiring more information beyond simply abstaining?" (Reviewer *Eiqe* and Reviewer *SnhN*)
- We present a novel human-annotated Directional Guidance Benchmark Dataset (Reviewer *SnhN*)
- Our paper introduces an effective synthetic data generation and training pipeline to improve the quality of MLLM/VLM responses. (Reviewer *eg4t*, Reviewer  *Eiqe*, and Reviewer *SnhN*)
---
Additionally, we would like to address some common concerns raised by the reviewers:

- **Concern 1**. Compared with more training-free methods (such as more sophisticated CoT and self-reflection prompting) for evaluating the baseline performance.

The prompt design is an open question, and we don't rule out that there will always be a better prompt design to get a performance boost.
We developed a series of prompting techniques to the best of our knowledge as mentioned in Section 4, and we will include more experiment results (including failure cases) in the revision. Meanwhile, we would like to highlight that the focus of this work is on the design of the task and the (data generation and training) pipeline that significantly improves the VLM's performance on this task.

- **Concern 2**. The task is narrowly scoped and limited.

While our study is based on a specific task, its implications and applications are wide-ranging. We are glad most reviewers emphasize the importance and necessity of this task (Reviewer *Eiqe* and *SnhN*) and hit one of the pain points of existing VLMs (Reviewer *Eiqe*). We also appreciate Reviewer *Eiqe* for highlighting that our task framing is straightforward yet has a potential real-world impact. Moreover, both the task and our proposed method can be generalized to other scenarios and potentially mitigate the struggles when models face insufficient information. Our contributions have the potential to make a significant impact on both the research community and practical deployments of VLMs, affirming the relevance and importance of our work in the broader AI landscape.

---

### Comment · Area_Chair_bGbP · 2024-08-12
**Please read the author rebuttal, other reviews and respond to the authors NOW!**

Dear Reviewers,

Thanks to those of you who already responded to the authors acknowledging the rebuttal and asking follow-up questions if any.

Those who have not responded yet, please do the following ASAP: thoroughly read the rebuttal, the other reviews and respond to the authors about whether all your questions / concerns have been addressed or not. If not, please elaborate on which questions / concerns are still not addressed so that the authors have fair chance of addressing them before the author-reviewer discussion period ends in ~41 hours from now (August 13th, 11:59pm AoE).

Your AC

---

### Decision · Program_Chairs · 2024-09-25

**Decision:**

Accept (poster)

**Comment:**

The reviewers find the problem being studied to be relevant and important, the proposed task framing to be simple and reasonable, the proposed data generation pipeline to be clever and the generated data to be novel and useful, and the performance gains to be impressive. The reviewers had raised some concerns, but the rebuttal successfully addressed most of them and all reviewers recommend acceptance. The authors are recommended to improve the final paper version by following the reviewer recommendations.